# Data Analysis of Impaired Renal and Cardiac Function Using a Combination of Standard Classifiers

**DOI:** 10.3390/jpm13030437

**Published:** 2023-02-28

**Authors:** Danijela Tasic, Drasko Furundzic, Katarina Djordjevic, Slobodanka Galovic, Zorica Dimitrijevic, Sonja Radenkovic

**Affiliations:** 1Clinic of Nephrology, UCC Nis, Medical Faculty, University of Nis, 18000 Nis, Serbia; 2Institute “Mihajlo Pupin”, University of Belgrade, 11060 Belgrade, Serbia; 3Vinca Institute of Nuclear Science—National Institute of the Republic of Serbia, University of Belgrade, 11000 Belgrade, Serbia

**Keywords:** kidney, heart, markers, machine learning, neural networks, forecasting ensembles, naive Bayes classifier, *k*-nearest neighbor

## Abstract

We examine the significance of the predictive potential of EPI cystatin C (EPI CysC) in combination with NTproBNP, sodium, and potassium in the evaluation of renal function in patients with cardiorenal syndrome using standard mathematical classification models from the domain of artificial intelligence. The criterion for the inclusion of subjects with combined impairment of heart and kidney function in the study was the presence of newly discovered or previously diagnosed clinically manifest cardiovascular disease and acute or chronic kidney disease in different stages of evolution. In this paper, five standard classifiers from the field of machine learning were used for the analysis of the obtained data: ensemble of neural networks (MLP), ensemble of *k*-nearest neighbors (*k*-NN) and naive Bayes classifier, decision tree, and a classifier based on logistic regression. The results showed that in MLP, *k*-NN, and naive Bayes, EPI CysC had the highest predictive potential. Thus, our approach with utility classifiers recognizes the essence of the disorder in patients with cardiorenal syndrome and facilitates the planning of further treatment.

## 1. Introduction

The combination of impaired kidney and heart function is present in life-threatening diseases that basically have mutually activated complex pathophysiological mechanisms. Through these mechanisms, the kidney and the heart directly regulate their function [1,2]. In clinical practice, kidney function is determined by formulas used to estimate the glomerular filtration rate (GFR) [3]. There are several equations for calculating the strength of glomerular filtration. Some of them are based on cystatin C values, of which the most commonly used is the CKD-EPI CyC (Chronic Kidney Disease Epidemiology Collaboration) [4,5]. A decrease in GFR occurs when about 50% of functional nephrons are lost, providing the capacity of the residual nephrons for filtration. GFR alone is not able to indicate the presence of conditions that affect kidney dysfunction. Therefore, a normal value of GFR in patients with cardiorenal syndrome can often be found despite the fact that there is an initial lesion on the kidneys [6,7,8]. Early detection of kidney damage is not possible with markers traditionally used in clinical practice [9]. Cystatin C is more useful for estimating GFR than the daily endogenous serum creatinine because it filters freely through the glomerulus, is reabsorbed completely, and is not secreted by the tubule cells in the kidneys. In clinical practice, it is suggested as a quick alternative marker for proving acute initial changes in the strength of glomerular filtration [10]. Cystatin C also provides numerous other informations, which is why it is important in distinguishing patients with a high degree of risk for complications and poor outcome in chronic renal failure [11]. In addition, it is useful for detecting patients who do not yet have manifested kidney disease, but have an increased risk of adverse cardiovascular events, heart failure, kidney damage, and death [12]. Cystatin C is not only a marker for the identification of high-risk renal and cardiovascular patients, but also a marker of cardiorenal syndrome (CRS) in the subclinical phase [13]. The use of cystatin C in daily clinical practice is limited due to its high cost, so according to KDIGO recommendations, it is used to assess renal function where serum creatinine values are unreliable [14]. In stable patients with chronic heart failure, numerous studies have shown that the accuracy of GFR calculated using the formula is relative [15]. Unstable clinical conditions such as acute heart failure, hypervolemia, intensive use of diuretics, and hemodynamic instability affect GFR values [16].

In clinical work, in addition to the assessment of kidney function, it is important to detect conditions that participate in the acute or chronic deterioration of kidney and heart function. For example, a control disorder in the regulation of electrolytes affects heart function due to an increased tendency of the appearance of arrhythmias, which are more common with changes in potassium levels [17]. The appearance of oscillations in electrolyte values is primarily influenced by medications, then impaired kidney function, then nutrition and patient compliance [18].

NTproBNP is a peptide used in daily clinical practice to predict heart failure, extracellular volume expansion, and cardiovascular events. It is a marker of outcome in patients with acute and chronic heart failure [19]. Although the level of NTproBNP, sodium, and potassium is somewhat dependent on kidney function, the combination of these biochemical parameters with EPI CysC may be significant for risk assessment, especially in the context of relatively preserved renal function in patients with cardiorenal syndrome [20,21].

Complicated pathophysiological mechanisms in cardiorenal syndrome require the use of a combination of biomarkers. Studies have shown that the predictive value differs depending on the level of a particular biomarker [22,23]. In addition to previously verified and precisely defined reference values by the manufacturer, the optimal cutoff values of biomarkers are constantly checked depending on the patient population being examined for the timely identification of patients who require early therapeutic intervention [24,25,26,27]. In such situations, it is important, based on reliable arguments, to distinguish those who are in a lifethreatening condition from those who do not require further examination or hospital treatment. In multiple pathophysiological mechanisms, it is difficult to automate the criterion of demarcation of positive and negative classes. That is why a statistical approach to the classification and interpretation of tests that patients undergo is increasingly being developed, which is based on data-driven classifiers, such as neuron networks or *k*-nearest neighbors [28]. The application of classification in medicine is performed by simple screening or tests that can be given in numerical (binary and continuous) or linguistic form or in certain classes (two or more) of subjects defined by labels. Multicriteri class discrimination is applied as a complex procedure characterized by a large number of features (predictors). The multidimensional feature space is used in the delimitation of different categories of respondents (positive and negative). They are most often used to form a database that contains, on the one hand, data on the mentioned characteristics and, on the other hand, the categorization (diagnosis) of patients. Features that are included in numerical or symbolic form in such databases, and which stand in a stronger or weaker direct correlation with the category of patients, incorporate useful information that is usually not visible at first glance, but can be extracted and used in prognostic or diagnostic purposes by various methods. [24]. Models used for the extraction of such knowledge are known as “data-driven predictors” and appear in the form of classifiers or in the form of function approximators (continuous values). When classifying data, we enter characteristics of patients (teaching input) and their corresponding output categories or classes of patients (positive and negative) and thus form a model of functional correspondence between them. By forming the parameters, the model becomes capable of generalizing the acquired knowledge to new cases and predicts the outcome of the test based on the feature data. Any classification process requires the selection of an operating point that will reflect a trade-off between the priorities required by the test, the biomarker, and the selected classifier. In serious health situations, the priority is to minimize FN (false negative) cases in order to focus attention on the subject’s condition. Economic criteria have the opposite tendency, avoiding unnecessary expensive diagnostic procedures and reducing the number of FP (false positive) cases. In practice, a compromise is usually made, which is always decided by medical experts.

Determining the cutoff value during the classification of samples can be performed through direct ROC analysis of the predictor variable (biomarker) using the cutoff value or using different classification models where the predictor variables (one or more) serve as input variables whose instance corresponds to previously assigned classes. Classifiers adapt their parameters in the direction of performance minimization, that is, the criterion of matching the predicted and actual belonging of instances to the current classes.

Each classifier, regardless of the complexity of the algorithm, can be replaced by a sufficiently large set of “If-Then” rules and classical logical reasoning tools, but in the case of a large number of predictor variables and a large number of instances, writing such a set of rules is practically unfeasible due to complexity. That is why, especially in medicine, standard classifiers from the field of artificial intelligence and fuzzy logic are used [25,26], which in a simpler way generate a hypersurface in a multidimensional space of predictor features that separates different classes of instances. Fuzzy logic was first described in detail by [29] and is based on a model of the human way of reasoning.

Today, a large number of databases in various scientific research areas are known and publicly available, such as the UCI Machine Learning Repository [30] and Center for Machine Learning and Intelligent Systems, and that number is constantly growing.

We conducted a pilot study with hospitalized subjects with cardiorenal syndrome using a method based on a mathematical model of classification from the domain of artificial intelligence.

The aim of our research is to examine the significance of the predictive potential of EPI cystatin C (EPI CysC) in combination with NTproBNP, sodium, and potassium in the evaluation of renal function in patients with cardiorenal syndrome using standard mathematical classification models from the domain of artificial intelligence or machine learning.

## 2. Methods

### 2.1. Patient Selection

The research was carried out at the Nephrology Clinic of the University Clinical Center in Nis and the Biochemistry Institute of the Faculty of Medicine in Nis. The research included 90 respondents, who were older than 18 years, of both sexes, who were divided into a clinical (*n* = 80) and a control group (*n* = 10). The respondents were familiar with the subject of the research and signed the informed consent. The research, as a cross-sectional comparative study, was approved by the Ethics Committee of the Faculty of Medicine in Nis (number 01-6481-9) and was conducted in compliance with the Declaration of Helsinki on medical ethics and the rules of good clinical practice. The control group included healthy individuals of both sexes with similar age and gender characteristics as the clinical group. The basic criterion for the inclusion of subjects with combined impairment of heart and kidney function in the study was the presence of newly discovered or previously diagnosed clinically manifest cardiovascular disease and acute or chronic kidney disease in different stages of evolution.

### 2.2. Laboratory Analysis and Data Collection

Basic clinical, functional biochemical, and hematological parameters were determined for all subjects, which included anthropometric profile, assessment of global kidney function, determination of functional glomerular reserve, and functional status of the cardiovascular system.

The sampled blood was centrifuged for 15 min at 1000/rev. Electrolyte values of sodium (Na) and potassium (K) were measured in serum on a Roche 9181 analyzer.

The concentration of BNP was determined on an Architect Abbott apparatus at the Center for Medical Biochemistry, University Hospital Nis.

Cystatin C (DSCTC0) was determined in plasma by an immunoturbidimetric method using commercial ELISA kits, R&D Systems, Quantikine, Abingdon, United Kingdom.

### 2.3. Equation Estimating GFR 

The determination of global kidney function (GFR) was estimated based on the value of cystatin C in the serum and was performed using the EPI cystatin C reference formula using a calculator [31]. GFR was calculated and adjusted to body surface area.

### 2.4. Statistical Analysis

For the assessment of the significance in the difference (*p*) between the tested values measured in two groups of subjects, the *t*-test or Mann–Whitney’s *U* test were used, dependent on the data distribution

## 3. Computer-Based Classification Using Standard Classifiers

Classification is generally the process of grouping entities into different categories according to discriminative attributes. Classifiers are le inductive learning predictors that establish a flexible functional correspondence between feature vectors of concrete instances and categories (classes) to which they belong. It is an incomplete induction that, on the basis of a limited representative sample, generates a conclusion about instances of the entire population that is not always a true judgment, in contrast to a complete induction that, on the other hand, is inapplicable in practice due to the limited available information resources. The classification algorithm implies the determination of a set of rules, that is, parameters in a general sense, during the training process, which assigns to each element of the set of *n* vectors ***x******i*** of the representative training sample *X* (***x******i*** ∈ *X*, *i* = 1,2, …, *n*) the corresponding element y*j* of the set Y of the *m* class (y*j* ∈, *j* = 1,2 …, *m*). Trained classifiers acquire the ability to generalize the classification process beyond the domain of the training sample to the complete domain of the target population. This means that a good classifier can extrapolate the acquired knowledge beyond the domain of the training sample with a high degree of confidence. The degree of reliability of the selected classifier is largely determined by the degree of representativeness of the training set, so one of the main prerequisites for determining a reliable classifier is a good selection of a representative training sample from the available set of instances. Instances are defined by a vector of discriminative features and, essentially, from the point of view of the classifier; they represent a point in a multidimensional feature space, so the representativeness of the observed training sample is conditioned by the nature of the distribution of these points in the feature space [28]. It should be mentioned that feature vectors can appear in several forms: binary, categorical, ordinal, integer, real, and combined. As a result of the training process, the following outcomes of the quality of the classifier are possible: (a) A bad identification phase of the model, which is manifested by both the bad classification of the training sample and the test sample. (b) A good identification phase of the model, which is manifested by a good classification of the training set and a good generalization confirmed on the test sample. In this case, the classifier’s error values on the entire training sample are small, so the model is considered reliable. (c) An apparently good identification phase of the model is manifested by excellent classification of the training set, which results in poor generalization, that is, poor classification of the test population. This phenomenon is known as model overfitting, which means overtraining or uncontrolled specialization to recognize only training instances [32].

## 4. Stable and Unstable Predictors

There are two categories of classifiers in terms of their stability: unstable and stable. An unstable predictor is one that has a stochastic nature, that is, an observable dependence on a randomly selected training set, so the hypothesis it forms during the test depends to a large extent on the selected domain to which the training set belongs. Examples of unstable predictors are decision trees and artificial neural networks. A stable predictor is one that does not have such strong dependencies on the training data. Examples of stable predictors are classifiers based on *k*-nearest neighbors and Fisher’s linear discriminator [33]. Unstable classifiers are known for a characteristically high variance of the generalization error signal, while stable classifiers have a low variance [34]. It is a feature that indicates the meaning of creating an ensemble of classifiers as one of the current topics in research. From this aspect, the idea of forming an ensemble of such classifiers in order to balance the compromise decision making of the predictors is completely understandable. Variance is only one component of the well-known decomposition of generalization error into two components, bias and variance [35]. These two values usually collide with each other: trying to reduce the bias component causes the variance to increase and vice versa. Bias is a measure of the difference between predicted and target values, and variance is a measure of the stability of the solution. A slight difference in the training data of a high-variance estimator will tend to produce huge differences in performance. Long-term predictor training reduces the bias, but gradually increases the variance so that, at some point, an optimal bias–variance ratio is achieved that minimizes the generalization error. This is the so-called current bias–variance dilemma [36,37].

## 5. Representativeness and Generalization

### 5.1. Representativeness of the Samples

A representative sample can be intuitively defined as an unbiased indicator of the state of the target population. A high degree of representativeness of the training sample implies a high degree of generalization accuracy of the relevant trained classifier, and therefore, the correct selection of the training sample is a key moment in the modeling of the selected process. According to [38,39,40], a representative sample is a carefully designed subset of the original data set (population), with three main properties: the subset is significantly reduced in terms of size compared with the original source set, and the subset better covers the main features from the original source than other subsets of the same size, and has as little redundancy as possible among the instances it contains.

### 5.2. Forecasting Ensembles

The design of the prediction process often relies on heuristic methods to improve model performance since no prediction method is perfect.

One of those methods is the combination of predictions of different predictor models from the same or different classes. This technique is also known as ensemble forecasting. 

There is an explanation of why ensemble algorithms outperform standard machine learning algorithms in both classification and regression:

Robustness: Ensembles are more resistant to noise and random deviations in the data, which is a good basis, for it can lead to more stable and reliable forecasting.

Diversity: By combining the hypotheses of individual models as ensembles, they are able to identify a wider range of distinctive patterns in the available data sample.

Overfitting reduction: By averaging the predictions obtained from multiple individual models, ensembles reduce the tendency of individual models to favor the data from the training set, resulting in improved generalization to the test data sample.

Increased accuracy: Ensembles have been proven in practice to outperform traditional machine learning algorithms in a variety of circumstances.

There are many techniques for creating forecasting models, including classical approaches such as ARIMA or machine learning methods such as decision trees or artificial neural networks. Combining multiple models leads to more accurate predictions based on predictor diversity [41]. One reason for this is that the combination of models reduces the possibility of choosing the wrong model. Nevertheless, it should be pointed out that the relationship between diversity and generalization is not fully understood and is still an open field for research [42], but this does not relativize the verified fact about ensemble efficiency.

The determination of the ensemble takes place in three phases:(a)Creation of a set of models: Construction of different individual members of the ensemble.

Diversity among models is the most important requirement to satisfy. Each model should generate acceptable but different hypotheses (predictions) from the others. A high degree of correlation between model predictions reduces the effectiveness of the ensemble and defeats the very purpose of the ensemble.

There are two main strategies for achieving the desired diversity of model behavior when building ensembles.

Combining different learning methods (heterogeneous ensembles) or combining the architecture and complexity of one learning method (homogeneous ensembles);Manipulation of the available training data set.

By repeating the random selection of the training set, different hypotheses are generated, and the required model diversity is achieved. A very effective and widespread method for manipulating the training set is ***Bootstrap aggregation*** (***bagging***) [43], which is described in more detail in a next subsection.

(b)Reduction: Removing weak and redundant models:

During the creation phase, a large number of models are created under stochastic conditions, but there is no guarantee that all models will be usable or increase the desired diversity of the ensemble. That is why the reduction of the number of models is carried out with the aim of preserving the diversity and performance of the ensemble, which is achieved by selecting models with the highest degree of internal noncorrelation and removing redundant models and models of low accuracy. This step increases the accuracy, robustness, and economy of the ensemble.

(c)Integration: combining selected models:

The last stage is integration and involves the combination of predictions of individual models.

The simplest way of integration is to take the average of the predictions of all individual models as the relevant answer, while in the case of classification the answer is formed by a majority vote of all models. An alternative solution is to assign different weights to each model and find a weighted average of the responses. Determining the significance or weight of a model’s response is based on the quality of its performance. A model with better performance obtains more weight.

### 5.3. Bootstrap Aggregation

The technique known in abbreviated form as bagging or “perturb and combine” (P&C) [43] is a meta-algorithm for training predictors, intended to improve the stability and accuracy of machine learning algorithms used in statistical classification and regression. It is also used as an effective tool for eliminating the influence of imbalances in the number of different classes, that is, the favoring of dominant classes during training and prediction. Bagging is one of the models created on the principle of averaging the performance of an ensemble of models trained on different subsets of the training sample. This algorithm is known and also reduces the variance of the error signal and helps prevent overfitting, that is, the tendency to specialize in better recognition of instances of the training sample compared with the test sample. Although originally applied to decision trees, it can be used without distinction with other predictors. Recent results in machine learning show that the performance of the final model should be improved not by choosing the structure of the best expected predictive model, but by creating a model based on the composition of the results (response) of models that have different structures. The reason is that, in fact, any hypothesis is only an estimate of the actual target value, and like any score, it is affected by bias and variance. Theoretical results show that variance reduction can be obtained simply by combining uncorrelated hypotheses about real target values. This simple idea is the basis of one of the most effective recent techniques in machine learning. Bagging leads to improved stability of classification models [34], such as artificial neural networks, decision tree classifiers, and subset selection in linear regression [44].

## 6. Applied Classifiers

The following text shows the basics of the five standard classifiers from the field of machine learning used for the analysis of the available data: ensemble of neural networks (MLP), ensemble of *k*-nearest neighbors (K-NN) and naive Bayes classifier, decision trees, and a classifier based on logistic regression.

### 6.1. Multilayer Perceptron 

Multilayer perceptron (MLP) belongs to neural networks with feed-forward signal propagation, the training of which takes place under supervision, that is, in the presence of a signal of the desired response to a predefined input. The training set is given in the form of a finite set of input–output pairs. MLP uses the generalized delta learning rule—the error signal back-propagation rule [45]. The perceptron is a structure organized into layers: input, hidden, and output layers. Information in the network moves from left to right. The nodes in the hidden and output layers calculate their activation value, which is weighted by weighting factors and transformed via the activation function into the output value. There are several different activation functions, of which the hyperbolic tangent is used by input and hidden layers and the identical mapping function used by output layers. 

The MLP ensemble is a very robust classifier [46]. Individual MLP structures usually converge to local minima, while the MLP ensemble converges to optimal solution. Individual MLP classifiers are also very sensitive to the imbalance of training samples. Therefore, MLP ensembles are used in order to reduce these negative phenomena. For the available database, we created an MLP ensemble in the form of 50 individual MLP structures. We designed the ensemble using individual MLP structures of varying complexities obtained by randomly selecting different numbers of neurons in the hidden layers. We used the Levenberg–Marquardt algorithm with an adaptive momentum to ensure an efficient training. We achieved overfitting control of the models by randomly selected sets of training and validation data. Every MLP ensemble classifier decides on the basis of majority voting criteria.

### 6.2. k-Nearest Neighbors Classifier

*k*-NN classifiers classify test instances based on the Euclidean distance in the feature space of a previously fixed number of nearest neighbors from the training set, which in this sense serves as a reference set of examples. Therefore, if the majority of the given numbers of neighbors from the training set that are closest to the selected test instance belong to class Ci, then that test instance belongs to the same class Ci. On the other hand, the classifier composed of the *k*-NN ensemble forms its answer based on the majority vote of each of the individual models. The applied *k*-NN ensemble consists of the following individual classifiers: 1-NN, 3-NN, 5-NN, 7-NN, and 9-NN. Due to the majority decision making of the ensemble, here we also used an odd number (5) of classifiers in the ensemble. It should be noted that *k*-NN classifiers are deterministic structures that always provide unique solutions for the same set of samples. The only stochastic influence on their response is the random selection of training and testing samples. This choice of training set increases the diversity of hypotheses of the classifier, which manifests itself by increasing the robustness and accuracy of the classification. It should also be noted that *k*-NN classifiers have a faster response compared with MLP structures.

### 6.3. Naive Bayes Classifier 

Like all other classifiers, the simple Bayes classifier implies the existence of a training set S of instances defined in the form of a feature vector ***x*** = (*x*1, …, *x**j*, …, *x**n*) of length *n* and their corresponding (associated) categorical or numerical class indicators *c**i* as elements of the set ***c*** = {*c*1, …, *c**i*, …, *c**L*} of cardinality |***c***| = *L*. In the training phase, the parameters of the model are formed, which later attach to each test instance ***x*** the estimated values of the probability of belonging to the given classes *P*^(*c**i*|***x***), i.e., (*P*^(***x*** ∈ *c**i*), *i* = 1, …, *L*) and based on the maximum of probability values, the final classification of test instances is performed. This classifier is actually an applied Bayes theorem (Thomas Bayes, 1701–1761), which is directly based on the calculation of the known Bayesian conditional probabilities.

### 6.4. Decision Trees

Decision trees are a widespread way of presenting the decision-making procedure through a branched structure in the form of a tree. It is often used to plan business and operational decisions in the form of a graphic flowchart. Decision trees are one of the most commonly used algorithms in machine learning. A decision tree algorithm is used to separate the features of a data set via a cost function. The optimization of a decision tree in purpose to eliminate branches that use irrelevant features is known as pruning. By adjusting the depth parameter of the decision tree, the risk of overloading or the complexity of the algorithm can be reduced. Decision trees in machine learning are mainly used for problems of classification or categorization of instances according to learned features. A big advantage of the decision tree compared with other machine learning algorithms is its simplicity; it is easy for visual presentation and understanding [47].

### 6.5. Logistic Regression

Logistic regression establishes a mathematical correspondence between a categorically defined target variable and one or more independent predictor variables. It is used in classification situations where the outcome for target variables is given in binary form while predictor variables can be given in continuous or categorical form [48]. This method is also used to identify important factors (*xi*) that determine the target variable (*y*), and also identifies the nature of the correspondence between each of these features and the target variable. The method uses a logistic function to form a hypothesis about the class membership of instances. Each instance feature has a corresponding weight whose determination takes place during the learning process in a way that minimizes the prediction error. Like other machine learning methods, logistic regression belongs to data-driven methods that are sensitive to the imbalance in the number of instances of different classes.

## 7. Performance and Metrics of Classifiers 

Performance results of individual predictors with defined cutoff values that serve as criteria for class membership are also presented. All these models (data-driven learning models) form parameters based on training pairs of input vector values of predictor variables *X* (features) and output values *Y* (labels or classes) of instances according to the following equation: *Y* = *T*(*X*), where *T* represents the operator for acceptable transformation of the input values of the training set into corresponding indicators of different classes. 

The characteristics of the classifiers are presented as a function of confusion matrices (Table 1) and their statistical derivatives, area under the ROC curve, sensitivity, specificity, and Youden’s index which are described in detail in the next chapter.

Let us consider a classification process using two classes where each instance *x* is mapped to a positive (*p*) or negative (*n*) element of the set *y* = {*p*, *n*} of class labels. A classifier performs a procedure of mapping from instances to predicted classes. Some classification predictors give a continuous output as an estimate of class membership probability for involved instances where different thresholds (values between 0 and 1 or 0% and 100%) can be used to predict class membership. One of these kinds of classifiers is multilayer perceptron (MLP) [49]. Other classifiers give a discrete class label. Total accuracy of classifications as one of the evaluation criteria is not suitable for the evaluation of unbalanced data [50]. Therefore, we used metrics to estimate the corresponding relative operational performance (ROC) curve [51,52,53,54], which is obvious in the example of classification of an unbalanced data set in which the dominant class (negative) contains 99% of available instances, while the minor class (positive) contains only 1% of available instances, which is often the case in practice. Using the metrics of total classification accuracy, in the above unbalanced set, we can declare the entire available group of patients as negative, absolutely ignoring positive cases, while maintaining the accuracy of classification at almost an ideal desired value of 99%. In this way, such an extremely high accuracy classifier will make a 100% error in recognizing positive cases, the recognition of which is essentially the goal of classification. Thus, all positive cases will be presented as false negative (FN). Accuracy is an appropriate metric of evaluation for classification only in case of well-balanced classes. 

We used the confusion matrix (Table 1) to define the displayed metric parameters when evaluating the performance of classifiers from unbalanced data sets as follows:
*Sensitivity* (true positive rate): *T**P**R* = *SEN* = *T**P*/*P* = *T**P*/(*T**P* + *F**N*)(1)

This quantity represents the sensitivity of the classifier in terms of the reliability of the correct classification of instances of the positive class. An ideal classifier will have a value of TPR = 1, which means that the number of correctly classified positive instances, true-positive cases (*TP*), is equal to the number of all positive (*P*) cases, *TP* = *P*, which also means that the number of positive cases falsely classified as negative, false-negative cases, is equal to zero (*FN* = 0) “Equation (1)”. Special attention should be paid to this size because, in certain cases, it can decide the outcome of the entire treatment of the patient.
*Specificity* (true-negative rate): *TNR* = *S**P**C* = *T**N*/*N* = *T**N*/(*T**N* + *F**P*)(2)

This quantity represents the sensitivity of the classifier in terms of the reliability of the correct classification of instances of the negative class. An ideal classifier will have the value *S**P**C* = 1, which means the number of correctly classified negative instances “Equation (2)”. True negative (*TN*) is equal to the number of all negative (*N*) cases, *TN* = *N*, which also means that the number of false-positive cases is zero (*FP* = 0) “Equation (3)”. In the case of a large number of FPs, the number of subjects on whom treatment should be applied (invasive follow-up examinations) increases, which results in an increase in costs but also in an increase in the positive outcome of patient treatment.
*False-positive rate*: *F**P**R* = *F**P*/*N* = *F**P*/(*F**P* + *T**N*) = 1 – *S**P**C*(3)

A receiver operating characteristics (ROC) graph is a technique for visualizing, organizing, and selecting classifiers based on their performance [53].

The ROC curve shows the parametric composition of *TPR*(*T*) values as a function of *FPR*(*T*), where *T* is a variable parameter that represents the value of the demarcation threshold between the distribution density functions of the observed classes. By choosing a certain point on the ROC curve, the values of sensitivity and specificity of the classifier are determined, which means its tendency to minimize the number of FN or to reduce the number of FP subjects in accordance with the criteria set by the biomarker designer or classifier model. An ideal classifier is characterized by the following values of the selected metric: *TP* = *P*, *TN* = *N*, *FN* = 0, *FP* = 0, *SPC* = 1, *TPR* = 1, *TNR* = 1, *FPR* = 0: These parameters are defined by the point A(0, 1) on the ROC diagram. A more detailed analysis of the ROC curve is given in [52].

The AUC parameter represents the value of the area under the ROC curve, and the performance of the classifier is directly proportional to this value, which in the case of an ideal classifier, that is, the correct classification of all instances, is 1. Other useful derivatives of the confusion matrix shown below are also often used in a more detailed analysis of the performance of the classifier “Equation (4)”.
*Youden’s index* = *Sensitivity* + *Specificity* − 1(4)

The validity of the classification is presented through the following categories: precision, recall and F-score, “Equation (5)”.
*Prec* = *TP*/(*TP* + *FP*); *Recall* = *TP*/(*TP* + *FN*); F-score = 2 × (Prec × Racall)/(Prec + Recall)(5)

## 8. Feature Subset Selection

An example of ROC analysis of a one-dimensional predictor is shown in Figure 1a,b with the accompanying text.

Figure 1 shows the predictor variable *X*(*x*1, *x*2, *x*3,…,*xn*) whose components xi are associated with the labels (+) and (–) as representatives of the positive (*P*) and negative (*N*) classes of instances. The purpose of the classifier is to establish a mathematical correspondence during training between all available realizations xi of the predictor variable *X* or realizations (*xi*, *yi*, *zi*,…) of a set of predictor variables (*X*, *Y*, *Z*,…) and the corresponding labels *P* and *N*, and to successfully classify new unknown cases based on the parameters of the model acquired during training. In the case of Figure 1a, a simple classification criterion is introduced, cutoff values (c1 and c2) that represent a simple border of classes *P* and *N*, so that all instances xi that are located to the left of c1 (first case) or c2 (second case) will be declared positive (*P*) and instances xi on the right will be negative (*N*). In the case of threshold c1, on the left, we have seven positive instances (+), which are classified as positive (*TP* = 7), and three negative instances, which are classified as positive (*FP* = 3). To the right of the threshold c1 is *TN* = 5 and *FN* = 5. In the case of the cutoff value c2, we have the following situation: *TP* = 8, *FP* = 5, *TN* = 3, and *FN* = 4. From these values, the values for specificity are calculated for both cases and sensitivity and select one of the cutoff values according to the adopted criterion. As you can see in the picture, you can define more cutoff values ci where *i* = 1,2,3,...20. Based on an integrative procedure for all ci values, the ROC curve is defined as a representative of the classifier’s performance.

Looking at Figure 1b, we conclude that it is an ideal nonlinear classifier with performance *TP* = *P*, *TN* = *N*, *FP* = 0, and *FN* = 0, where *P* = 12 (number + labels) and *N* = 8 (number–labels), which corresponds to the ideal point A(1,1) on the ROC curve. The given example shows the case of a one-dimensional predictor for the sake of easier understanding of the problem, but the same criteria apply to the multidimensional predictor space.

The general conclusion from the example in Figure 1 is the fact that linear classifiers like cutoff values or linear perceptron have worse performance compared with nonlinear classifiers like MLP, *k*-NN, or decision trees and that they can be seriously counted on as data-driven models in the case of a solid set of available data. An MLP classifier was the first neural network able to solve the XOR problem, providing nonlinear decision boundary [45,49].

## 9. Results

### 9.1. Study Population

The clinical group consisted of patients, with an average age of 70.72 ± 9.26 years. After comparing, serum electrolytes (Na+, K+) did not differ significantly in the clinical group from those in the control group. The tested biomarkers showed significantly higher values in the clinical group than in the control group: BNP (*p* < 0.001), cystatin C (*p* < 0.001). The parameters of the functional status of the renal system of the clinical and control groups, with significant differences obtained, were the ones calculated with the EPI formula (*p* < 0.001), as shown in Table 2.

In our paper, four relevant parameters were analyzed in clinical and control group subjects—EPI cysC, NTproBNP, Na, and K—as predictors in two ways: (a) using each of the parameters individually (Table 3, Table 4, Table 5 and Table 6) and (b) using a combination of all four parameters (Table 7).

### 9.2. Training Data

The results obtained by testing the performance of six different classifiers using the complete available of patient data are presented. During the estimation of the classifier parameters, we used a set of randomly selected 50% majority class examples and 50% minority class examples for training, keeping the existing imbalance rate. Model testing was performed on the remaining 50% of the available data set.

The *training data* is divided into a *learning set* of 85% and a *validation set* of 15% of the *training data*, which serves to prevent overfitting of the MLP.

Each model from the ensemble receives a different random set for training and validation in accordance with the *bagging* method. In this way, the impact of class imbalances is also reduced. 

All results are presented numerically and graphically in the relevant tables and figures. Table 3, Table 4, Table 5 and Table 6 show the results of all predictor variables individually (EPI cysC, NTproBNP, Na, and K), while Table 7 shows the results of the combination of all predictor variables as a common input vector. In the first column of all tables, the names of the parameters of the used metrics are listed, while the classification methods are listed in the second column. In the second column of Table 3, Table 4, Table 5 and Table 6, the cutoff values of individual predictors are given, and the performance parameters are defined based on them. Each subsequent column refers to the performance of the specified classifiers.

**Table 3 jpm-13-00437-t003:** The diagnostic performance of t six classifiers for CRS by using a single predictive variable: GFR of biomarker EPI cysC (mL/min/1.73 m^2^). Metrics of performance are: AUC, specificity, sensitivity and Youden’s index.

Predictor	Single Biomarker: EPI cysC (mL/min/1.73 m^2^)					
Classifiers/ Metrics	Cutoff = 116.000	MLP Ensemble	*k*-NN Ensemble	Naive Bayes	Logistic Regression	Decision trees
AUC	0.9094	**0.9643**	0.9556	0.9556	0.9094	0.8887
Specificity	0.9625	0.9626	0.8000	0.8000	0.8000	0.9000
Sensitivity	0.8000	0.8000	0.9375	0.9375	09125	0.9750
Youden’s index	0.7625	0.7626	0.7375	0.7375	0.7125	0.8750

**Table 4 jpm-13-00437-t004:** The diagnostic performance of the six classifiers for predicting CRS by using single predictive variable: biomarker NTproBNP (pg/mL). Metrics of performance are: AUC, specificity, sensitivity, and Youden’s index.

Predictor	Single Biomarker: NTproBNP (pg/mL)					
Classifiers/ Metrics	Cutoff = 26.000	MLP Ensemble	*k*-NN Ensemble	Naive Bayes	Logistic Regression	Decision trees
AUC	0.9013	0.9413	0.9406	0.6000	0.9413	**0.9800**
Specificity	0.8000	0.8000	0.8000	0.2000	0.9000	0.9000
Sensitivity	0.7000	0.9000	0.9000	1.0000	0.8875	0.9500
Youden’s index	0.5000	0.7000	0.7000	0.2000	0.7875	0.8500

**Table 5 jpm-13-00437-t005:** The diagnostic performance of six classifiers for predicting CRS by using single predictive variable: sodium concentration Na (mmol/L). Metrics of performance are: AUC, Specificity, Sensitivity and Youden’s index.

Predictor	Single Biomarker: Na (mmol/L)					
Classifiers/ Metrics	Cutoff = 138.000	MLP Ensemble	*k*-NN Ensemble	Naive Bayes	Logistic Regression	Decision trees
AUC	0.5663	0.8625	0.7143	0.5000	0.5663	0.9313
Specificity	0.6375	0.8000	0.7000	0.0000	0.5000	0.8000
Sensitivity	0.5000	0.8250	0.7125	1.0000	0.6375	0.8750
Youden’s index	0.1375	0.6250	0.4125	0.0000	1375	0.6750

**Table 6 jpm-13-00437-t006:** The diagnostic performance of six classifiers for predicting CRS by using single predictive variable: potassium concentration K (mmol/L). Metrics of performance are: AUC, specificity, sensitivity, and Youden’s index.

Predictor	Single Biomarker: K (mmol/L)					
Classifiers/ Metrics	Cutoff = 4.8000	MLP Ensemble	k-NN Ensemble	Naive Bayes	Logistic Regression	Decision trees
AUC	0.6244	0.8481	0.6475	0.5000	0.6244	**0.8818**
Specificity	0.5000	0.8000	0.4000	0.0000	0.5000	0.8000
Sensitivity	0.5625	0.7875	0.8875	1.0000	0.5625	0.8125
Youden’s index	0.0625	0.5875	0.2875	0.0000	0.0625	0.6125

**Table 7 jpm-13-00437-t007:** The diagnostic performance of five classifiers for predicting CRS by using combination of four predictors EPI cysC (mL/min/1.73 m^2^), NTproBNP (pg/mL), Na (mmol/L), and K (mmol/L). Metrics of performance are: specificity, sensitivity, and Youden’s index.

Predictors	Combined Biomarkers: NTproBNP (pg/mL), EPI cysC (ml/min/1.73 m^2^), Na (mmol/L), K (mmol/L)				
Classifiers/ Metrics	MLP Ensemble	*k*-NN Ensemble	Naive Bayes	Logistic Regression	Decision trees
AUC	0.9937	0.9887	0.8187	0.9863	**0.9981**
Specificity	1.0000	1.0000	0.7000	0.9000	1.0000
Sensitivity	0.9875	0.9625	0.9375	0.9500	0.9625
Youden’s index	0.9875	0.9625	0.6375	0.8500	0.9625

In Table 7, there is no cutoff column because combination of all predictors were used as inputs. That is why in Table 7, we have only five columns with the values of the classifiers’ performance parameters.

An ideal classifier would receive a score equal to 1.000 for all displayed values of the performance parameters both on the available test sample and on the entire population from which the available frame instance was selected. It should be said that the ideal classifier is unattainable for almost all practical problems, especially for the domain of medicine. From the results, we conclude that the decision trees and MLP classifiers have the best score, both in the case of individual predictors (Table 3, Table 4, Table 5 and Table 6) and in the case of a combination of all predictors (Table 7).

It can also be observed that performance of classifiers trained on a combined set of predictors is obviously better than the performance of classifiers that used individual predictor variables.

## 10. Graphic Representation of the Performance of the Applied Classifiers

It should be noted that when determining the characteristic point on the ROC curve, we tried to choose the one with compromise values of sensitivity and specificity, which is visible in all the images shown.

The biomarker EPI cysC has the greatest prognostic significance (AUC = 0.9094) Figure 2a. The diagonal line refers to the classifier with random class selection.

The highest single prognostic significance has the biomarker EPI cysC (AUC = 0.9643). The classifier based on the combination of all examined markers Figure 3e has almost ideal performance (AUC = 0.9937 and specificity = 1). The diagonal line refers to the random classifier and has no prognostic significance.

The biomarker NTproBNP Figure 4b (AUC = 0.9800) has the greatest single prognostic significance. The classifier based on the combination of all predictors has almost ideal performance (AUC = 0.9981 and specificity = 1). The diagonal line refers to the random classifier and has no prognostic significance.

The greatest single prognostic significance has the biomarker EPI cysC Figure 5a (AUC = 0.9556). The classifier based on the combination of all markers has almost ideal performance Figure 5e (AUC = 0.9887 and specificity = 1).

The greatest single prognostic significance has the biomarker EPI cysC Figure 6a (AUC = 0.8562). The classifier based on the combination of all examined markers has a weaker performance Figure 6e (AUC = 0.8187) compared with the other classifiers. The variables Na and K Figure 6c,d have no prognostic significance in the case of the Bayes classifier and behave identically to the random classifier.

The highest single prognostic significance has the biomarker NT proBNP Figure 7b (AUC = 0.9413). The classifier based on the combination of all predictors has good performance Figure 7e (AUC = 0.9863 and specificity = 0.9000). The predictors Na and K Figure 7c,d have very weak prognostic significance in the case of this classifier and behave similarly to a random classifier (AUC < 0.6000).

The result was obtained by extrapolating (generalizing) the classification criteria to the homogeneous feature space using the MLP classifier. In Figure 8b,c, there is an obvious tendency that low EPI cysC values correspond to the clinical group with cardiorenal syndrome (CRS), while higher values are characteristic of the control group. Medically justified correlations of other variables, NTproBNP, Na, and K, with the respective patient groups were also applied. A more comprehensive presentation of the distribution of patient groups as a function of the three predictors (NTproBNP, Na, and EPI CysC) is given in Figure 8d. MLP classifiers and those based on fuzzy logic are suitable for generalization in multidimensional space.

## 11. Discussion

Medicine develops thanks to the adoption and monitoring of novelties that occur in other scientific disciplines and their application in the diagnosis, prevention, and treatment of various diseases [55]. Until now, the traditional approach to diagnosing and treating diseases is “one size fits all” and has proven to be unreliable and unpredictable [56]. That is why an individual approach to the treatment of patients according to their needs and characteristics is on trend [57,58]. In order to replace the simplified approach with a more precise one, predictive, precision medicine (PPM) was introduced, which combines new fields using biomarkers to predict the onset of the disease, its progression, and its treatment adapted to individual characteristics [59,60,61,62,63,64,65,66,67]. In order to distinguish patients according to their risk for kidney disease and distinguish them in relation to their potential in cardiorenal syndrome, in this paper, we applied a scientific technique in which computers solve the problem without programming using better algorithms and the most commonly used different monitoring techniques from the domain of machine learning using standard classifiers. Ensemble of neural networks (MLP), ensemble of *k*-nearest neighbors (*k*-NN) and naive Bayes classifier, decision tree, and a classifier based on logistic regression. The classifier based on logistic regression is a parametric, discriminative, fast, and simple method for classifying independent variables in relation to the dependent variable. Unlike it, naive Bayes is useful in small independent sets. Unlike logistic regression, *k*-nearest neighbor (*k*-NN) is slower, supports nonlinear solutions, and cannot derive the confidence level. Decision tree requires no preprocessing of the data, is efficient in terms of collinearity, and provides high purity of predictions by pruning the tree. Compared with naive Bayes, it is a discriminating, easier, and more flexible model. Both models find nonlinear solutions and allow for interaction between independent variables.

The goal of modern medicine is a personalized approach that is based on individual variability in the mechanism of disease occurrence and risk factors. This very important goal in the field of cardiology and nephrology to gain insight into disease mechanisms in daily practice is achieved using markers for clinical scenarios. It is very difficult to distinguish reversible organ damage in cardiorenal syndrome, which is why our discussion is focused on the possibilities of a personalized diagnostic approach in cardiorenal syndrome using available markers.

The aim of this paper was to explain the predictive potential of the markers and propose new methods as an aid in clinical decision making. Our investigation allows us to determine which of the used variables is the most helpful for detecting the primary impact of altered kidney function in patients with cardiorenal syndrome using the given patterns and selecting the most relevant data in the entire set. The results showed that in MLP, *k*-NN, and naive Bayes, EPI cysC had the highest predictive potential, while the decision tree and the classifier based on logistic regression suggested that NTproBNP was superior in terms of predictive effect compared with the other examined variables. The classifiers made it possible to distinguish the essence of the disorder in patients with cardiorenal syndrome and facilitate the planning of further treatment.

The basic concept of cardiorenal syndrome is based on common pathophysiological mechanisms that cause simultaneous disturbances in the function of both organs, and over time, this convergent pathophysiological complex causes damage to both organs. Recently, the opinion is that the dominant pathophysiological disorder in the clinical entities of cardiorenal syndrome can be distinguished by means of biomarkers in order to find a therapeutic solution more easily. However, an increase in serum creatinine or other parameters of kidney function should be evaluated within the clinical picture, because for the application of the appropriate dose and type of drugs in the therapy of heart failure, the difference between real and pseudo kidney damage is important [68,69,70]. Thus, the authors of the DOSE study (Diuretic Optimization Strategies Evaluation) showed, paradoxically, that although the subjects recovered their kidney function, the patients were re-hospitalized because the length of the previous treatment was shorter, and the effect of eliminating excess water was incomplete [71]. An increase in nitrogen products and a weakening of renal function are present during the introduction of neurohormonal antagonists [72]. Studies examining drugs for heart failure, such as inhibitors of angiotensin-converting enzyme, mineralocorticoid receptors, diuretics, and beta blockers, randomized selected patients to drugs with the tested active substance and to placebo [73,74,75,76]. Renal function in cardiorenal syndrome can be a limiting factor for the application of many drugs including mechanical cardiac support and heart transplantation [77,78].

The fundamental principle of machine learning that was used in our work was not applied in the so far published research on the application of artificial intelligence tested in research with the clinical entity of cardiorenal syndrome. Our proposed paradigm on the analysis of renal function in cardiorenal syndrome is inspired by previous clinical research in this area, but the application of standard classifiers provides an original contribution in this area.

## 12. Conclusions

In this work, we have shown that it is possible to identify different subgroups of patients with cardiorenal syndrome using biomarkers that are available in everyday clinical practice. It is a small contribution to the future clinical approach to ensure “the right treatment for the right patient at the right time”.

## Figures and Tables

**Figure 1 jpm-13-00437-f001:**
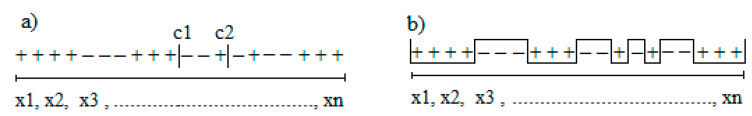
Decision boundary between positive (+) and negative (−) class of instances. Simple linear cutoff values (c1 and c2) defined in one-dimensional feature space (**a**) and nonlinear boundary defined by MLP neural network classifier (**b**).

**Figure 2 jpm-13-00437-f002:**
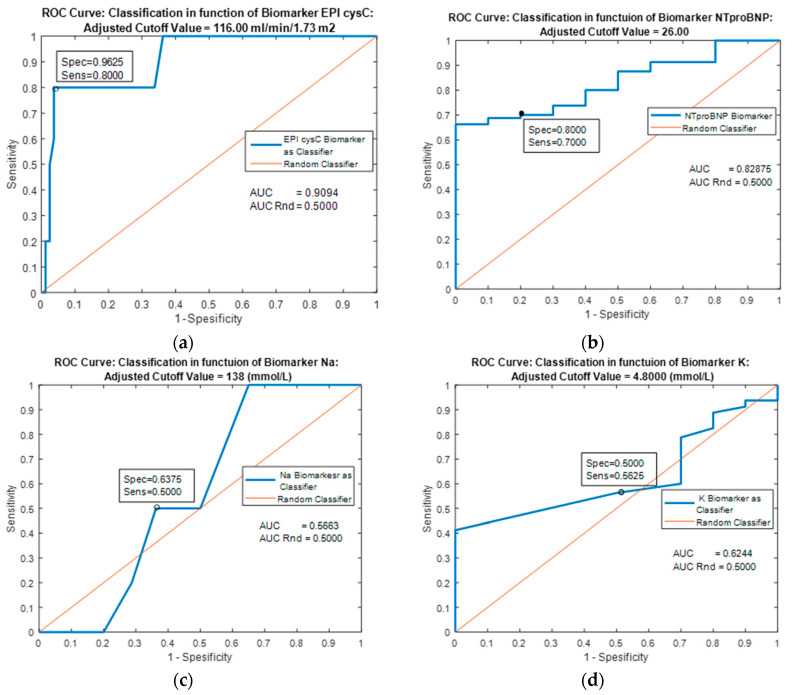
Performance comparison of classifiers based on four individual predictors using their variable cutoff values as classification criteria.

**Figure 3 jpm-13-00437-f003:**
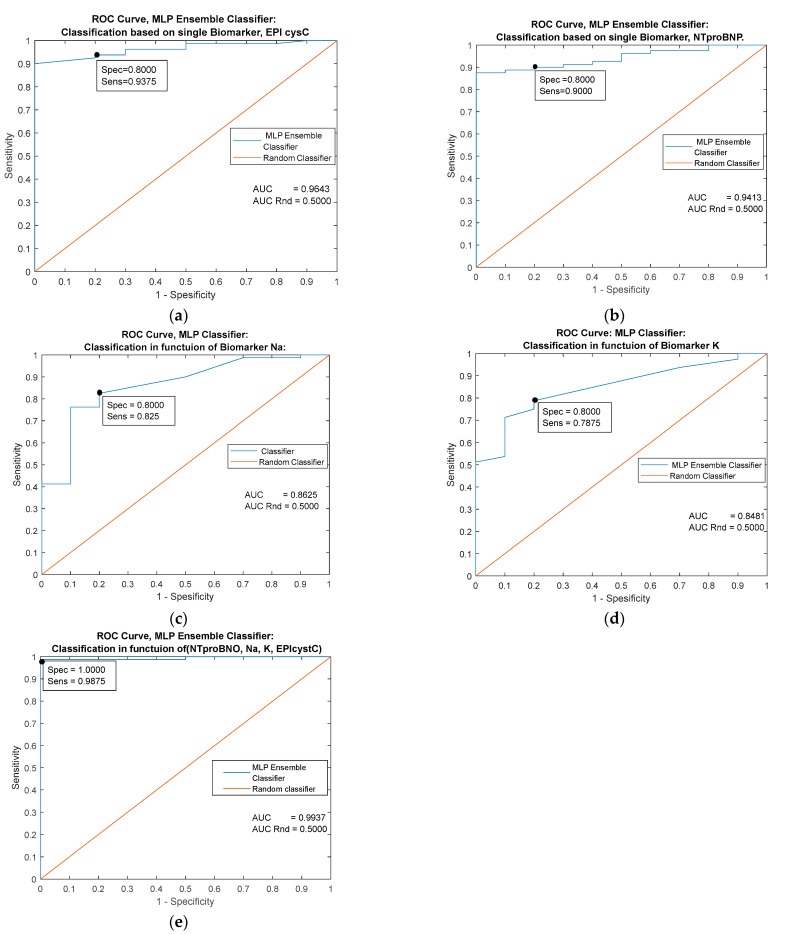
Performance comparison of MLP classifiers based on four individual predictors (**a**–**d**) and on the combination of all the mentioned predictors (**e**).

**Figure 4 jpm-13-00437-f004:**
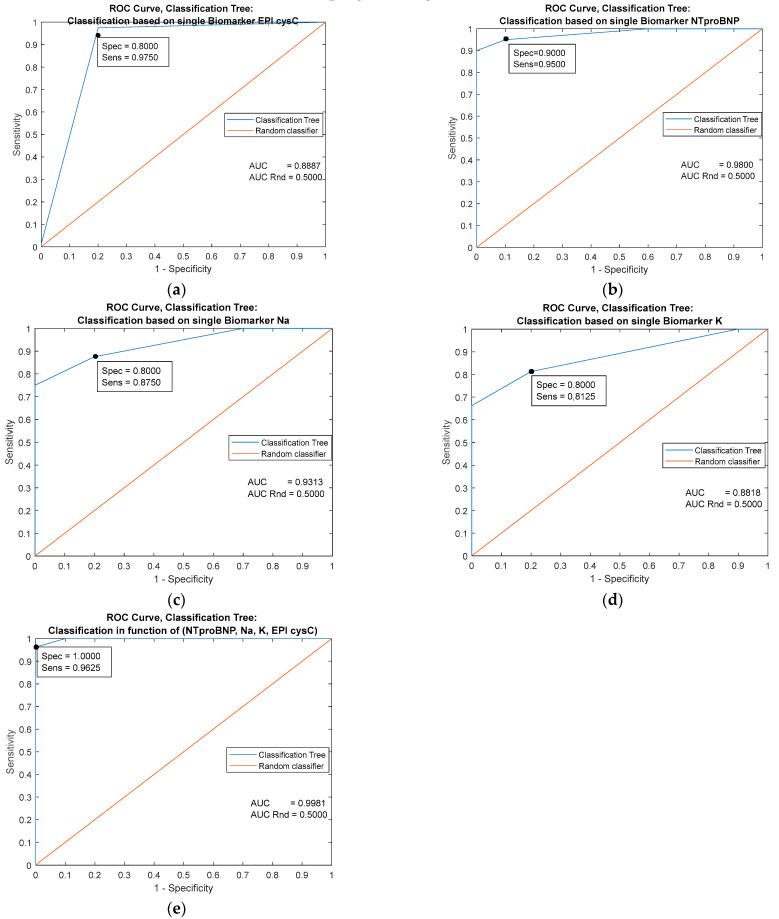
Comparison of the performance of decision tree classifiers based on four individual predictors (**a**–**d**) and on the combination of all mentioned predictors (**e**).

**Figure 5 jpm-13-00437-f005:**
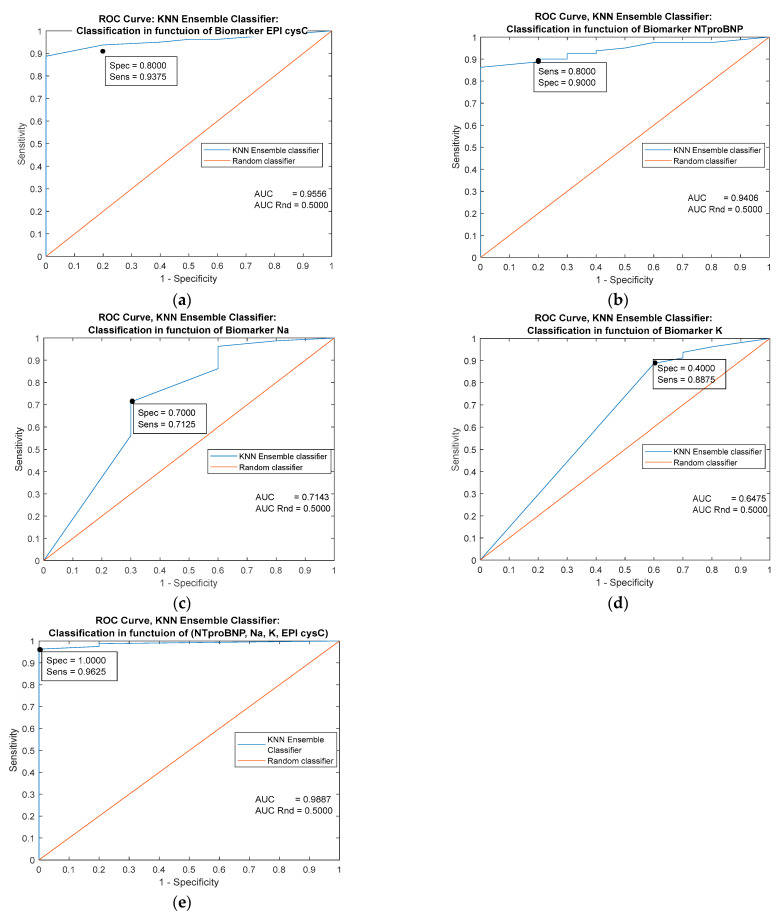
Performance comparison of *k*-NN ensemble classifiers based on four individual predictors (**a**–**d**) and on the combination of all the mentioned predictors (**e**).

**Figure 6 jpm-13-00437-f006:**
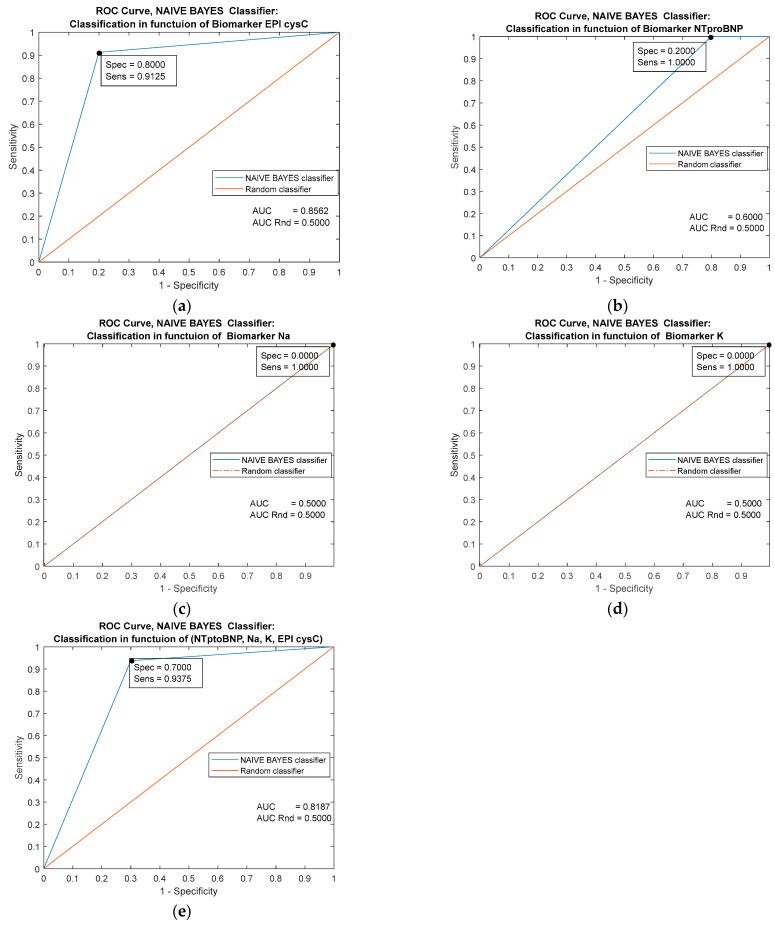
Performance comparison of the Naive Bayes Classifiers trained on four individual predictors (**a**–**d**) and on the combination of all mentioned predictors (**e**).

**Figure 7 jpm-13-00437-f007:**
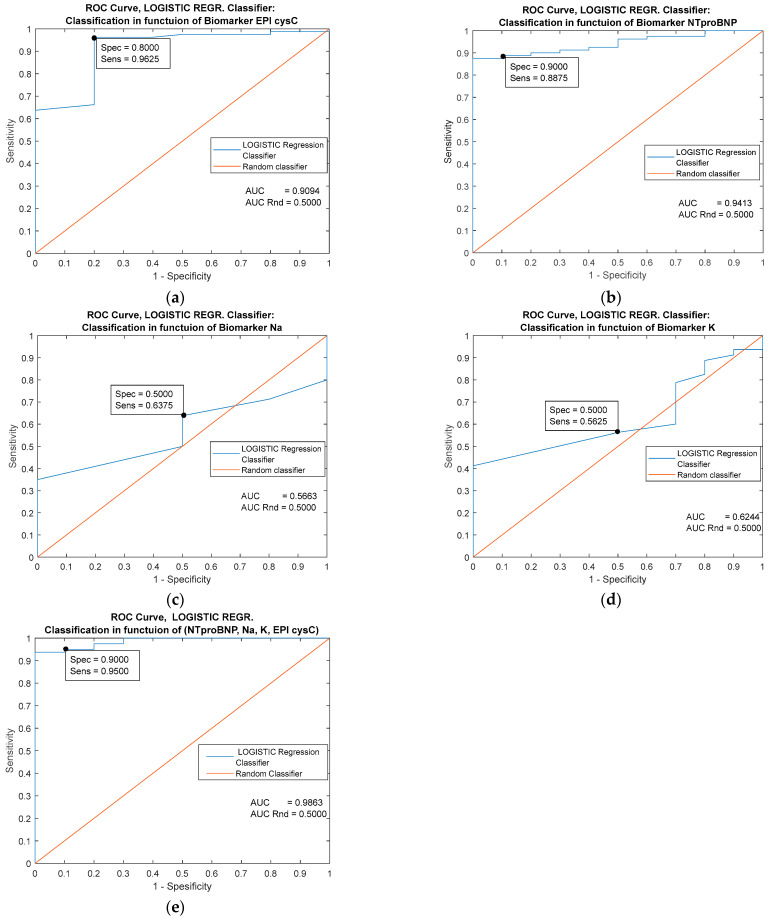
Performance comparison of logistic regression classifiers based on four individual predictors (**a**–**d**) and on the combination of all the mentioned predictors (**e**).

**Figure 8 jpm-13-00437-f008:**
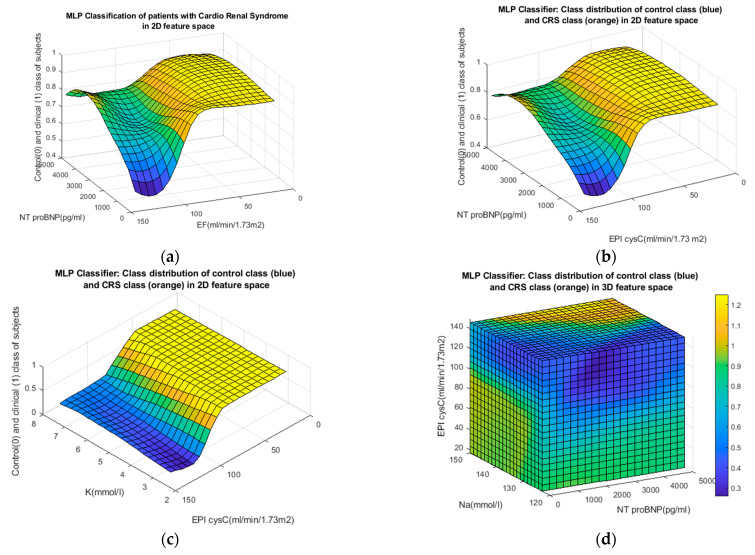
Distribution of patient groups, control (blue) and clinical (orange) in 2D feature space (**a**–**c**) and 3D feature space (**d**).

**Table 1 jpm-13-00437-t001:** Confusion matrix.

		Real Class	
		*p*	*n*
Predicted class	*Y*	*TP* (true positive)	*FN* (false negative)
	*N*	*FP* (false positive)	*TN* (true negative)

*P*: Number of positive realizations, where from the aspect of classification: *P* = *TP* + *FN*; *N*: Number of negative realizations, where from the aspect of classification: *N* = *TN* + *FP*; *TP*: Number of positive cases classified as positive; *TN*: Number of negative cases classified as negative; FP: Number of negative cases classified as positive; *FN*: Number of positive cases classified as negative.

**Table 2 jpm-13-00437-t002:** Demographic values and values of laboratory parameters of patients in the clinical and control groups.

Parameters	Clinical Group	Control Group	*p*/*Z*	*p*
Age (year)	70.72 ± 9.26	69.55 ± 32.01	1.079	0.286
Na (mmol/L)	137.75 ± 4.77	139.67 ± 0.82	1.513	0.130
K (mmol/L)	4.89 ± 0.99	4.27 ± 0.45	1.870	0.061
EPI cystatin C (mL/min/1.73 m^2^)	40.43 ± 25.15	105.5 ± 14.32	10.332	<0.001
BNP(pg/mL)	1451.23 ± 1591.39	19.43 ± 9.18	4.890	<0.001
Cystatin C (mg/L)	1.91 ± 0.73	0.50 ± 0.15	5.141	<0.001

*t*-test, Mann–Whitney *U* test.

## Data Availability

Correspondence and requests for material should be addressed to D.T.

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
