# Peer review of "Data Analysis of Impaired Renal and Cardiac Function Using a Combination of Standard Classifiers"

_jpm, 2023, doi:10.3390/jpm13030437_

Round 1
Reviewer 1 Report
The authors applied machine learning to predict the disease. The idea is interesting. However, their model needs further optimization and verification.
Major:
1. All machine learning algorithms use default parameters, but different parameters are needed for different data; different machine learning algorithms are suitable for different applications.
2. The samples should be divided into trained and tested groups to improve the robustness of the results.
3. The discussion section should include at least the similarities and differences with other studies. e.g., Another group reported that the serum electrolytes (Na+, K+) were significantly different between disease and control groups.
4. Generally, the AUC > 0.7 had clinical implicants. Further effort should be devoted to modifying and optimizing these models.
Minor
1. There are indirect references and formatting errors in the references.
2. Cardiorenal syndrome is a complex disease, and the authors should provide detailed clinical data on the patient.
Author Response
Comments for manuscript ID 2104920
Dear Editor,
And Journal of Personalized Medicine Editorial Office,
We are greatful for the editor s and referees suggestions for improvement our manuscript „ Data analysis of impaired renal and cardiac function using a combination of standard classifiers” by Danijela Tasić, Draško Furundžić, Katarina Đorđević, Slobodanka Galović, Zorica Dimitrijević, Sonja Radenković.
We corrected the manuscript according to reviewers suggestions.
All corrections are presented in red in the revised version of manuscript.
In addition the list of changes and the answers to the reviewers questions are given below.
To Reviewer 1
Major:
- All machine learning algorithms use default parameters, but different parameters are needed for different data; different machine learning algorithms are suitable for different applications.
Response: We would like to thank the reviewer for carefully reading and commenting on our work.
Each of the applied algorithms belongs to data driven (dependent) algorithms, which means that the determination of their parameters takes place during the training process under the important influence of the representativeness of the sample. The first criterion by which we compare the used ML algorithms (classifiers) is the identical set of data on which they were trained, and the second criterion is the same performance presentation system of the ML classifier.
The goal is to find a better performing classifier that increases the reliability of the conclusions. Due to the stochastic nature of algorithms, it is difficult to find their optimal structure and parameters, but through data resampling, the level of their performance increases.
Different algorithms are more suitable for different types of data, but during the selection of a set of standard classifiers, we counted on this incompatibility and through the methodology of selection of training instances and interpretation of results, we made this incompatibility acceptable from our point of view.
Remarks from point 1 resulted in an additional detailed explanation of the methodology applied in the paper, which is presented on new pages of the text in subsections 2.1, 2.2, 2.3, 2.4.
- The samples should be divided into trained and tested groups to improve the robustness of the results.
Response: We would like to thank the reviewer for carefully reading and commenting on our work.
We thank you for the careful observation of the deficiency regarding the more detailed presentation of the training sample manipulation procedure, even though we applied it. We fixed this shortcoming in the subsection Training data of chapter 4.
- The discussion section should include at least the similarities and differences with other studies e.g. Another group reported that the serum electrolytes (Na+,K+) were significantly different between disease and control groups.
Response: We would like to thank the reviewer for carefully reading and commenting on our work.
„Thus, the authors of the DOSE study ( Diuretic Optimization Strategies Evaluation) showed, paradoxically, that although the subjects recovered their kidney function, the patients were re-hospitalized because the length of the previous treatment was shorter, and the effect of eliminating excess water was incomplete. An increase in nitrogen products and a weakening of renal function is present during the introduction of neurohormonal antagonists. Studies examining drugs for heart failure such as inhibitors of angiotensin-converting enzyme, mineralocorticoid receptors, diuretics, beta blockers randomized selected patients to drugs with the tested active substance and to placebo. Renal function in cardiorenal syndrome can be a limiting factor for the application of many drugs including mechanical cardiac support and heart transplantation.
The fundamental principle of machine learning that was used in our work was not applied in the so far published research on the application of artificial intelligence tested in research with the clinical entity of cardiorenal syndrome. Our proposed paradigm on the analysis of renal function in cardiorenal syndrome is inspired by previous clinical research in this area, but the application of standard classifiers provides an original contribution in this area.”
- Generally, the AUC > 0.7 had clinical implicants. Further effort should be devoted to modifying and optimizing these models.
Response: We would like to thank the reviewer for carefully reading and commenting on our work.
We agree with your point of view and any performance improvement is precisely aimed at optimization and software implementation of new marker technologies that include all available relevant data. Raising the performance level aims to increase the reliability and accuracy of the hybrid methods used.
Minor 1. There are indirect references and formatting errors in the references.
Response: We would like to thank the reviewer for carefully reading and commenting on our work.
We have removed the observed indirect references. Thank you for your careful observation of impermanence.
- cardiorenal syndrome is a complex disease and the autors should provide detailed clinical data on the patients.
Response: We would like to thank the reviewer for carefully reading and commenting on our work.
„ The basic concept of cardiorenal syndrome is based on common pathophysiological mechanisms that cause simultaneous disturbances in the function of both organs, and over time this convergent pathophysiological complex causes damage to both organs. Recently, the opinion is that the dominant pathophysiological disorder in the clinical entities of cardiorenal syndrome can be distinguished by means of biomarkers in order to find a therapeutic solution more easily. However, an increase in serum creatinine or other parameters of kidney function should be evaluated within the clinical picture, because for the application of the appropriate dose and type of drugs in the therapy of heart failure, the difference between real and pseudo kidney damage is important.”
Thank you very much.
With respect and gratitude,
Danijela D Tasić
danijeladt@gmail.com
University of Niš, Medical Faculty,
UCC Niš, Clinic of Nephrology, Serbia
Katarina Đorđević
Slobodanka Galović
Draško Furundžić
Zorica Dimitrijević
Sonja Radenković
Reviewer 2 Report
Title can be reframed
Keywords should be precise to explain the title of the article
Even though the results shown are attracting the readers to concentrate a little explanatory discussion on each result attained is appreciated to make even the budding researchers understand the concept of the article.
When the methods of Machine learning are dealt with, a formal introduction to them is required.
State of the art related works are needed. Do not include general works that exist in the domain. Include references that are related to the ML applications to the domain of article.
Author Response
Dear Editor,
And Journal of Personalized Medicine Editorial Office,
We are grateful for the editor's and referees' suggestions for the improvement of our manuscript „ Data analysis of impaired renal and cardiac function using a combination of standard classifiers” by Danijela Tasić, Draško Furundžić, Katarina Đorđević, Slobodanka Galović, Zorica Dimitrijević, Sonja Radenković.
We corrected the manuscript according to the reviewer's suggestions.
All corrections are presented in red in the revised version of the manuscript.
In addition, the list of changes and the answers to the reviewer's questions are given below.
To REVIEWER 2:
- Title can be reframed
Response: We would like to thank the reviewer for carefully reading and commenting on our work.
„Data analysis of impaired renal and cardiac function using a combination of standard classifiers”
2.Key word should be precise to explain the title of the article
Response: We would like to thank the reviewer for carefully reading and commenting on our work.
classifiers, kidney, heart, markers, machine learning, neural networks, forecasting enseembles, naive bayes classifier, k-neareest neighbour
3.Even thought the results shown are attracting the readers to concentrate a little explanatory discussion on each result attained is appreciated to make even the buddin researchers understand the concept of the article.
Response: We would like to thank the reviewer for carefully reading and commenting on our work.
„The goal of modern medicine is a personalized approach that is based on individual variability in the mechanism of disease occurrence and risk factors. This very important goal in the field of cardiology and nephrology to gain insight into disease mechanisms in daily practice is achieved using markers for clinical scenarios. It is very difficult to distinguish reversible organ damage in cardiorenal syndrome, which is why our discussion is focused on the possibilities of a personalized diagnostic approach in cardiorenal syndrome using available markers.”
„ The basic concept of cardiorenal syndrome is based on common pathophysiological mechanisms that cause simultaneous disturbances in the function of both organs, and over time this convergent pathophysiological complex causes damage to both organs. Recently, the opinion is that the dominant pathophysiological disorder in the clinical entities of cardiorenal syndrome can be distinguished by means of biomarkers in order to find a therapeutic solution more easily. However, an increase in serum creatinine or other parameters of kidney function should be evaluated within the clinical picture, because for the application of the appropriate dose and type of drugs in the therapy of heart failure, the difference between real and pseudo kidney damage is important. Thus, the authors of the DOSE study ( Diuretic Optimization Strategies Evaluation) showed, paradoxically, that although the subjects recovered their kidney function, the patients were re-hospitalized because the length of the previous treatment was shorter, and the effect of eliminating excess water was incomplete. An increase in nitrogen products and a weakening of renal function is present during the introduction of neurohormonal antagonists. Studies examining drugs for heart failure such as inhibitors of angiotensin-converting enzyme, mineralocorticoid receptors, diuretics, beta blockers randomized selected patients to drugs with the tested active substance and to placebo. Renal function in cardiorenal syndrome can be a limiting factor for the application of many drugs including mechanical cardiac support and heart transplantation.
The fundamental principle of machine learning that was used in our work was not applied in the so far published research on the application of artificial intelligence tested in research with the clinical entity of cardiorenal syndrome. Our proposed paradigm on the analysis of renal function in cardiorenal syndrome is inspired by previous clinical research in this area, but the application of standard classifiers provides an original contribution in this area.”
When the methods of Machine learning are dealt with, a formal introduction to them is required.
Response: We would like to thank the reviewer for carefully reading and commenting on our work.
At your request, in section 2.4 we have corrected a carefully observed shortcoming and added appropriate text related to the characteristics of the applied ML methods.
- State of the art related works are needed. Do not include general works that exist in the domain. Iclude references that are related to the ML applications to the domain of article.
Response: We would like to thank the reviewer for carefully reading and commenting on our work.
In the presented text, we tried to provide relevant references that support the essence of the analyzed problem. When we reduce the set of cited works to only those that deal with the key problem of this research as well as similar applied methods, then the choice is quite modest. We tried to fulfil all the requirements as best as possible in order to eliminate major shortcomings.
The authors answered all comments and suggestions.
Finally, I would like to thank once again the referees for their valuable comments and suggestions, which improved our manuscript.
Thank you very much.
With respect and gratitude,
Danijela D Tasić
danijeladt@gmail.com
University of Niš, Medical Faculty,
UCC Niš, Clinic of Nephrology, Serbia
Katarina Đorđević
Slobodanka Galović
Draško Furundžić
Zorica Dimitrijević
Sonja Radenković